# Mechanical and Electronic Video Stabilization Strategy of Mortars with Trajectory Correction Fuze Based on Infrared Image Sensor

**DOI:** 10.3390/s20092461

**Published:** 2020-04-26

**Authors:** Cong Zhang, Dongguang Li

**Affiliations:** Science and Technology on Electromechanical Dynamic Control Laboratory, Beijing Institute of Technology, Beijing 100081, China

**Keywords:** trajectory correction fuze, infrared image sensor, video stabilization, gray projection

## Abstract

For a higher attack accuracy of projectiles, a novel mechanical and electronic video stabilization strategy is proposed for trajectory correction fuze. In this design, the complexity of sensors and actuators were reduced. To cope with complex combat environments, an infrared image sensor was used to provide video output. Following the introduction of the fuze’s workflow, the limitation of sensors for mechanical video stabilization on fuze was proposed. Particularly, the parameters of the infrared image sensor that strapdown with fuze were calculated. Then, the transformation relation between the projectile’s motion and the shaky video was investigated so that the electronic video stabilization method could be determined. Correspondingly, a novel method of dividing sub-blocks by adaptive global gray threshold was proposed for the image pre-processing. In addition, the gray projection algorithm was used to estimate the global motion vector by calculating the correlation between the curves of the adjacent frames. An example simulation and experiment were implemented to verify the effectiveness of this strategy. The results illustrated that the proposed algorithm significantly reduced the computational cost without affecting the accuracy of the motion estimation. This research provides theoretical and experimental basis for the intelligent application of sensor systems on fuze.

## 1. Introduction

Trajectory correction fuze is opening a low cost and high profit way to improve attack accuracy for various projectiles [1]. By installing a fuze, projectiles can obtain the correction function without adding any sensors or changing the size. However, with increasingly complex operational backgrounds and missions, the guidance mode of trajectory correction fuze faces greater challenges. In this case, the infrared image sensor is widely used because of its long detection distance, good anti-interference ability and strong concealment. Therefore, the installation of a fuze with an infrared image sensor can not only improve attack accuracy, but also meet different mission requirements, which has received much attention. Li [2,3,4] proposed a novel trajectory correction fuze based on an image sensor. Then, the correction strategy of mortars was studied to improve the attack accuracy. However, the previous research was based on the macroscopic trajectory and ignores the effect of projectile’s jitter on the image. 

Compared with missiles, the infrared image sensor is rarely used in mortars. Since mortars are often launched from a smooth bore gun, they have a certain amount of micro-spin to keep balance. This causes a rotation of the field on the infrared image sensor. In addition, due to the influence of aerodynamic force, mortar will produce high frequency jitter. The above reasons cause the instability of the image, resulting in fuze not being able to accurately detect the target. Therefore, keeping a stable field is the precondition of improving the attack accuracy. We want to make a graceful integration of the video stabilization and fuze that finally benefits the efficiency of mortar. 

Generally speaking, video stabilization technology has been widely used in military and civilian fields, such as investigation, fire control, surveillance, in-vehicle and handheld cameras [5]. It can be roughly classified into three categories, namely mechanical image stabilization (MIS) [6], optical image stabilization (OIS) [7] and electronic image stabilization (EIS) [8]. Compared with MIS and EIS, the components of OIS are complex and expensive. Additionally, the mortar has a high overload during launch, which will destroy the optical structure and make OIS impossible. Some MIS methods use gyro-scope for motion estimation [9,10], but that can only capture the rotational motion, leaving the translational motion undetected. The above problems lead to a lesser application of video stabilization technology on fuze. Therefore, the inner components and space of the fuze should be reasonably coordinated. In addition, the fuze’s information is updated quickly due to the short trajectory distance and high-speed flight, resulting in a once stable cycle time that is much less than that of other types of carriers. Given the above, we designed an open-loop control and multiple-cycle strategy based on the magnetometer and stepper motor to reduce the complexity of the fuze.

Generally speaking, the EIS can be equivalent to video stabilization and belongs to the category of image processing. According to the motion model of the projectile, video stabilization can be divided into 3D [11,12,13] and 2D [14,15,16,17] approaches. Liu [18] applied the structure from the motion to the video frames and used the content-preserving warps for a novel view synthesis. Zhou [19] introduced 3D plane constraints for improved warping quality. All the 3D methods conducted expensive and brittle 3D reconstruction for stabilization. On the contrary, 2D methods use a series of 2D linear transformations for the motion estimation. Then, the smoothed motion is computed by using a Gaussian filter [20,21], a Kalman filter [22,23,24] or least-square fitting [25]. In previous research, the methods of feature matching [26,27,28,29,30] were usually used to estimate the motion vector. However, complex and dense features are difficult to establish in a real infrared ground image. Based on that, Wu [31] proposed a gray projection algorithm based on the Sobel gradient for infrared video. Even so, calculating the gradient of all the pixels will still increase the computational cost. In addition, both feature-matching and gradient calculation need a threshold, which is usually given directly through experience and is not adaptive. Consequently, traditional 2D methods can hardly maintain the balance between the robustness and computational cost. Thus, we proposed the adaptive global gray threshold to preprocess the infrared image so as to reduce the computational cost and increase the Peak Signal to Noise Ratio(PSNR). Then, two different median filters were used before and after estimating the motion vector to improve the efficiency. 

In this paper, improving the real-time and accuracy of video stabilization was the main purpose. We proposed a novel mechanical and electronic stabilization strategy for trajectory correction fuze on mortar, where the infrared image sensor was designed to provide a stable video output according to target characteristics and mission requirement. For the further improvement of the accuracy of the fuze’s roll angle detection, a small magnetometer sensor was used to replace the gyro-scope. The control actuators were reduced and the mechanical stabilization was completed only by a single-axis stepper motor. In order to choose a suitable stabilization algorithm, the transformation relation between the fuze jitter and pixel movement was investigated to establish the relationship between the motion of the projectile and the image. Additionally, an adaptive global gray threshold (AGGT) method was proposed for the image pre-processing. Based on that, the image was partitioned and the global motion vector was quickly estimated by gray projection. Importantly, the image stabilization ability of this method was compared with three other representative algorithms through the simulation experiment. Finally, three experiments were performed to verify the effectiveness of the mechanical and electronic image stabilization systems. 

This paper is organized as follows: in Section 2, the workflow of the MIS on fuze is introduced. The parameters of the target-based infrared image sensor are designed. The transformation relation between the image and the fuze is established. In Section 3, the AGGT method and gray projection algorithm are proposed for EIS. Then, the effectiveness of the strategy was verified by simulation experiment. In Section 4, three experiments were performed to verify the effectiveness of the system. The conclusion is presented in Section 5.

## 2. Mechanical Video Stabilization

### 2.1. General Workflow of the Fuze

As illustrated in Figure 1, the newly designed fuze is divided into three components, the forward part, mid part and aft part, respectively. The forward yellow part containing the infrared image sensor is strapdown with the fuze. The head of the fuze is the fairing, indicated in orange. The mid part which in blue can dependently rotate relatively to the aft part by the control of a motor. The magnetometer sensor is also included. The aerodynamic control force for the trajectory correction is generated by a pair of rudders, which are shown in green. The aft part is shown in purple, which is used to screw onto the projectile. The more detailed structures of the fuze are mentioned in reference [2], which mainly include the transmission and fixing method of the motor and gear. This mechanism is used to transmit the driving moment from the motor in aft fuze.

Figure 2 shows the workflow of the mechanical stabilization for the fuze. The system starts to work when the projectile passes over the apex of the trajectory. The initial roll angle of the projectile is random, and the aft part of the fuze has a 1–2 r/s micro-spin due to the moments of the tail wing. The actual roll angle of the projectile is measured through a magnetometer sensor, then the angle is transmitted to the projectile-borne computer. To ensure the infrared image sensor always remains stationary with respect to the geodetic coordinate system, by the motor’s control, the mid and forward part of the fuze rotate relatively to the projectile at the same speed. This workflow will be repeated many times until the projectile enters the terminal trajectory and the infrared image sensor starts to work.

The following two norms should be accomplished throughout the design and workflow of the fuze:The imaging plane of the infrared image sensor should be perpendicular to the longitudinal axis of the projectile, and the imaging plane center should be located on the longitudinal axis.The roll angle detected by the magnetometer sensor should not exceed two degrees. The response time of the system should not be more than one second. 

Obviously, the workflow of the mechanical stabilization can effectively reduce the influence of the projectile’s rotation on the video. With this design, the electronic video stabilization only needs to solve the problem of the video jitter without considering the rotation.

### 2.2. Design of Infrared Image Sensor Strapdown Fuze

To obtain continuous and stable video information, the infrared image sensor strapdown fuze was designed. In this section, a method for calculating the three main parameters of the infrared image sensor was proposed, and the design was completed.

The infrared image sensor consists of two parts, which are the lens assembly and the sensor engine. We choose a simple and stable coaxial spherical lens assembly due to the strong impact during the projectile’s launch. According to the miniaturization and thermal sensitivity requirements of the fuze, the uncooled staring VOx microbolometer was selected. Therefore, the focal length, array format and the field of view were needed to determine the performance of the infrared image sensor:(1)$$\frac{R}{f}=\frac{H}{h}$$
(2)$$Fov_{col}=2\arctan\left(\frac{nh}{2f}\right)$$
(3)$$Fov_{row}=2\arctan\left(\frac{mh}{2f}\right)$$

The relationship between the three parameters is expressed in Equations (1)–(3). Equation (1) is proposed to build a bridge between the target information and the fuze, in which *f* is the focal length of the sensor lens, *R* is the linear distance between the projectile and the target, *H* is the characteristic size of the target and *h* is the pixel size. It should be noted that the focal length can be regarded as the image distance, because the image distance is far less than the object distance. The relationship between the array format and the field of view is shown in Equations (2) and (3), in which *Fov_col_* and *Fov_row_* represent the vertical and horizontal fields of view, respectively, and *n* × *m* is the size of the array format on the sensor.

According to the terminal trajectory of the mortar, the detection distance and the field of view can be determined, which respectively is denoted as *R* and *Fov*. *H* and *h* are determined by the target size, where *H* could be obtained by literature, and *h* was calculated by Johnson criteria. Therefore, the value of *f* and *n* × *m* could be obtained. Figure 3 shows the composition of the designed infrared image system with the fuze. The blue bracket was used to strapdown the infrared image system with the forward of the fuze, which connected the sensor engine with two screws. The lens assembly could be screwed into the bracket. The experimental verification of the correctness of the infrared image sensor design is shown in Section 4.

### 2.3. Transformation Relation

In order to solve the jitter problem, we should investigate the change of the field of view caused by the pitch and the yaw motion of the fuze. In this section, the necessary coordinate system is introduced and the transformation relation between the fuze jitter and the pixel movement is established as a preliminary study.

Due to the symmetry of the coaxial spherical lens assembly, the complex motion of the infrared image sensor can be represented by analyzing the single direction motion. When the projectile shakes around its center of mass, the motion vector of the infrared image sensor can be divided into three directions, the translation along and perpendicular to the axis of the projectile, and its rotation. The definition of the sensor coordinate system o-xyz is as follows: the origin is located at centroid of the lens assembly, the x–o–y plane is parallel to the array format plane, and the z axis coincides with the projectile axis and points up. 

The relationship between the shaky sensor and the moving pixel is shown in Figure 4 and Figure 5, representing the translation and shake, respectively. The ellipses in the two figures express the lens assembly. The dotted line indicates the projectile axis. *f* and *H* represent the focal length and the detection distance. The displacement of the sensor is represented by Δ*s* and Δ*z*. The rotation angle is indicated by Δ*θ*. *d* and *d*′ indicate the position of the imaging plane at the start and end of the movement, respectively: (4)$$\Delta d_{1}=\frac{f}{H}\Delta s$$
(5)$$\Delta d_{2}=\frac{d\Delta z}{H-\Delta z}$$
(6)$$\Delta d_{3}=f\left(\tan\left(\theta +\Delta \theta \right)-\tan\theta \right)$$

Equations (4)–(6) show the distance of the pixel movement calculated by the triangle similarity during the integration time. Through approximate calculations, it can be concluded that the order of magnitude of Δ*d*_1_ and Δ*d*_2_ is much smaller than that of a single pixel, but the order of Δ*d*_3_ is approximately equal to the order of the pixels. Therefore, it is necessary to solve the influence of shaking on the movement of the pixel. In this paper, a method of fast electronic video stabilization with gray projection accurate to pixel-level was selected.

## 3. Electronic Video Stabilization

When there are moving dim objects in the scene, the traditional gray projection algorithm will reduce the accuracy of the global motion vector estimation due to the calculation of the sum of all gray values. In this section, the image is divided into blocks and quickly filtered to maintain the accuracy of the global motion estimation without increasing the computational cost.

### 3.1. AGGT 

The difference between infrared image and RGB image is that the infrared image’s background area has a low contrast, which has a serious effect to the estimation of global motion vectors during the image stabilization. Therefore, the adaptive histogram equalization method can increase the contrast of the background of the infrared image to highlight the difference of gray level. In addition, the image was divided into sub-blocks which were used to estimate global motion vectors. It should be noted that the number, size and density of the sub-blocks directly affected the accuracy and computation of the global motion estimation. The amount of pixel movement leading by the shakiness of the projectile should not exceed the size of the block, and the relationship between them can be calculated by Equation (6). The sub-blocks were arranged according to the array. Then, the even numbered blocks were taken for each row and column, so as to avoid double counting the same area and reduce the number of pixels which needed to be calculated by four times in theory:(7)$$col_{k}\left(j\right)=\overset{n}{\underset{j=1}{\sum }}G_{k}\left(i,j\right)$$
(8)$$rol_{k}\left(i\right)=\overset{m}{\underset{i=1}{\sum }}G_{k}\left(i,j\right)$$

After image partition, gray projection in horizontal and vertical directions was adopted on all sub-blocks to generate two independent one-dimensional curves. The calculation method is shown in Equations (7) and (8), where *col_k_*(*j*) and *row_k_*(*i*) represent the sum of the gray values of any sub-block in the frame *k*, *n* and *m* are the size of block and *G_k_*(*i*,*j*)is the gray value of the pixel at the (*i*,*j*) position.:(9)$$\varepsilon =\frac{m\times n}{row\times col}\overset{row}{\underset{i=1}{\sum }}\overset{col}{\underset{j=1}{\sum }}G_{k}\left(i,j\right)$$
(10)$$\max\left[col_{k}\left(j\right)\right]-\min\left[col_{k}\left(j\right)\right]\geq \varepsilon \left(m,n,\bar{G_{k}}\right)$$

In addition, due to the regions in some blocks, these may have unclear features and a uniform distribution of gray value. Therefore, sub-blocks which may lead to inaccurate motion estimation should be eliminated. First, a threshold was defined to evaluate the sub-block, which is represented in Equation (9). Second, when the largest D-value between the projection values of the row and the column in the sub-block was greater than *ε* at the same time, this sub-block was retained, otherwise, it was removed. As shown in Equations (9) and (10), row and col are the size of the image, and $\bar{G_{k}}$ is the average of the gray value. The value of *ε* was generated by large numbers of experimental derivations. Compared with the gradient threshold algorithm, this method did not need to calculate all the pixels, which greatly reduced the computational cost. 

Figure 6 shows experimental results taken from a frame of an infrared-image-guided missile hitting a target. Figure 6a shows the original image, whereas Figure 6b is a result after the adaptive histogram equalization. It can be clearly seen from the processed image that the contrast was greatly improved and the texture of the background was more obvious. Figure 6c shows the details of the partition and sifting, where the green areas satisfy the condition of Equation (10), otherwise, the red areas are ignored. The size of the sub-blocks is 48 × 48, determined by the calculation of the Equation (6). In order to embody the algorithm in detail, two typical neighboring sub-blocks were selected from the background to show the results of the projection curve. In the corresponding plots, the blue and brown curves represent the gray projection values in the horizontal and vertical directions, respectively. In the first plot, there were obvious max and min values in both curves, and the difference met the limiting conditions. In the second plot, although there was a significant difference in the horizontal gray projection value, the vertical value changed only a little, so this sub-block was ignored. The results showed that 11 of the 36 sub-blocks in the frame were selected, which saved 69.4% of the number of pixels compared with the unprocessed frame, in theory. In addition, the AGGT and gradient threshold methods were used for the partition and sifting of 100 images of the same video sequence. Among them, AGGT consumed an average of 13 ms and the gradient threshold method consumed an average of 75 ms. Compared with the gradient threshold algorithm, the AGGT reduced the computational cost of the partition and sifting by 82.7% under the same simulation conditions.

### 3.2. Motion Estimation, Filtering and Compensation

Selecting the correct reference frame is the first step in accurately estimating the global motion vectors. Since the projectile is always in flight, the inter-frame matching method was used. Specifically, the previous frame in the two adjacent frames was used as the reference frame, and the next frame was the current frame. Due to the flight time of the projectile not being more than 5s in the terminal trajectory, the stabilization method only worked in the early stage. When the frame rate was 25Hz according to the performance of the infrared sensor, the number of loaded images did not exceed 125. Additionally, the image stabilization period of the system was no more than 1s. Therefore, the accumulated error of the motion vector was small. Moreover, the high degree of information overlap between the two adjacent frames will simplify the difficulty of motion estimation: (11)$$C\left(w\right)=\overset{N}{\underset{j=1}{\sum }}{\left[col_{cur}\left(j+w-1\right)-col_{ref}\left(m+j\right)\right]}^{2},1\leq w\leq 2m+1$$
(12)$$d_{y}=m+1-w_{\min}$$

The size and direction of the motion vector were calculated from the row and column curves of the gray projection values in the two adjacent frames. The difference function of the y direction was calculated as shown in Equation (11), where *C*(*w*) represents the difference function between the column curve of two sub-blocks at the same position in the two adjacent frames. *Col_cur_* and *col_ref_* are the gray projection values of the *j*th column of the current frame and the reference frame, respectively. *N* is the length of each sub-block. The maximum range of the estimated motion vector is represented by *m*. *w*_min_ is the value of *w* when *C*(*w*) is the minimum value. Therefore, the displacement vector in the *y* direction is represented by d*y*, as shown in Equation (12). Similarly, the displacement vector in the *x* direction can be generated.

Notably, there was unevenly distributed noise in the real infrared scenes, which was caused by weather factors and the sensor itself. Therefore, the image needed to be filtered before calculating the sub-blocks. In addition, a motion vector could be obtained from each remaining sub-block. However, incorrect motion vectors were still generated because of the motion of small objects in the background. To solve these problems, two median filters *f*_1_ and *f*_2_ were proposed to process the candidates. *f*_1_ is a 3 × 3 filter to remove the noise from all the frames, and *f*_2_ selected the median of all motion vectors as the global motion vector. Figure 7 illustrates an example, where all the arrows denote the motion vectors. All the vectors were arranged in the order of direction after applying the median filter *f*_2_ and the incorrect vector (yellow arrow) was removed at the same time. Furthermore, the unique motion vector was selected as the global motion vector. The median filter was frequently used in feature estimation and was treated as the secret of a high-quality flow estimation. Here, the similar idea was borrowed for sparse motion regularization. 

After the unique global motion vector was determined, the current frame was compensated backwards according to the motion vector with the value of |dx,dy|. If d*y* was positive, it indicated that the current frame moved up |dy| pixels relatively to the reference frame. Otherwise, the current frame moved down relatively to the reference frame. The horizontal direction motion vector dx could be processed in the same way. Notably, in order not to affect the speed of online stabilization, Equation (11) does not need any future motions for optimization. However, the AGGT has a one frame latency since no matter how fast the optimization runs, it occupies some time.

### 3.3. Simulation

To compare our method with the previous methods, two special videos were prepared for the processes. The first video was the process from the launching to hitting a moving target of a certain type of infrared guided ground-to-ground missile. The other one was an air-to-ground missile hitting a static target. We ran our method on a laptop with 2.8 GHz CPU and 8G RAM. To evaluate the quality, two objective metrics were introduced: PSNR and computational cost. To demonstrate the superiority of our method, we also made comparisons with the method of the L1-norm, Meshflow and Sobel gray projection. L1-norm and Meshflow were two typical feature-matching algorithms. Figure 8 shows the testing results on the dataset. Here we counted the average of the PSNR and computational cost to compare the advantages and disadvantages between adaptive global gray threshold (AGGT) and other three algorithms comprehensively. 

The average PSNR of the L1, Meshflow, Sobel and AGGT in the first video were 32.1dB, 20.5dB, 33.5dB and 35.1dB, respectively. In the second video, the numbers were 27.3dB, 26.1dB, 30.2dB and 31.5dB, respectively. It can be seen that both the Sobel and AGGT have a higher value of the average PSNR than the L1 and the Meshflow. Especially in the second video, the average PSNR of the AGGT obtained a significant increment compared to the other three algorithms with 4.3dB, 5.5dB and 1.4dB respectively. Additionally, the computational cost of the AGGT obtained a great reduction compared to the L1 and the Meshflow in the first video. Even to the Sobel, it was also reduced by 18.3%. 

The above data showed that the difference in the PSNR between the Sobel and AGGT was very little. In order to further compare the stabilization details of the AGGT with the other algorithms, the trajectories of the PSNR in the two videos is shown in Figure 9. It can be seen that the PSNR of AGGT was larger than that of the other algorithms in both videos. We noticed that in this example, the first one suffered from a massive unpredictable large translation, while the second one had just a slight shakiness. Although Meshflow and L1 are relatively stable in Figure 9a, the low PSNR indicates that the stabilization results were seriously distorted. In contrast, the results of the AGGT were consistent with video shakiness. Consequently, the AGGT had a better capability and robustness than the other two algorithms which could greatly reduce the computational cost yet maintain a large value of PSNR in infrared video stabilization.

## 4. Experimental Analysis

In this part, three experiments were performed to verify the rationality of the infrared image sensor, mechanical stabilization and system stabilization, respectively. 

### 4.1. Experiment of Detection Capability of Infrared Image Sensor

The prerequisite for the subsequent experiments was that the infrared image sensor could clearly observe small targets at long distances. In this experiment, a vehicle model with an electrically conductive board was used to replace the infrared real target. The infrared image sensor was carried by an UAV to achieve equivalent long-range detection. The equivalent conditions and the parameters of sensor are shown in Table 1.

With experience, the actual size of the vehicle was 2.3 m. When the detection distance was 1.5 km, it should have been equivalent to 65 m in order to detect the model with a size of 0.1 m. Simultaneously, choosing the appropriate average outside temperature (298.15 K) ensured that the experiment met the actual conditions. Pixel size was determined by the detection distance and conformed to the current technology of the infrared sensor. The focal length and field of view (FOV) can be computed with the expressions Equations (1)–(3) in Section 2. The large array format ensured that the target did not exceed the field of view when the projectile was in the ideal pitch (53 degree).

The result of the equivalent detection is expressed in Figure 10. The small white area in the center of the field of view is the equivalent small target. By enlarging the area of the target, the number of the target’s pixels was more than what was originally set. This was due to the thermal radiation characteristics of the target. The results proved that the infrared image sensor could clearly detect the model within 1.5 km, and the design method was correct. 

### 4.2. Experiment of Mechanical and Electronic Stabilization

In order to evaluate the effects of the mechanical and electronic stabilization, a new method combining visual observation and target detection was used in this experiment. Considering the outdoor conditions, a turntable and location plate were used to reflect the influence of outdoor conditions, such as the detection distance and aerodynamic force. The distance between the turntable and the location plate was 5 m which was equivalent to Section 4.1. Moreover, aerodynamic force would increase the amplitude of the projectile’s swing, causing the trajectory to shift and the target to exceed the field of view. The experiment used a location plate with a diameter of 1.5 m and the FOV of the sensor (Table 1) so that the target did not exceed the maximum deflection range of the trajectory by ± 120 m (Reference 2). Additionally, under the combined action of the moment and the wind, the mortar jittered within ± 5 degrees. This was achieved by controlling the rotation range of the turntable. The azimuth of the target always changed within 10 degrees.

The efficiency of this method was evaluated by comparing the azimuth of the target before and after the stabilization. Figure 11 shows the experimental equipment and environment, in which Figure 11a shows the prototype of the fuze’s structure, Figure 11b is the 3-DOF turntable and Figure 11c is the location plate. The small infrared target was replaced by a thermistor and placed on the location plate after calculating the equivalent distance. First, the fuze and the thermistor were powered on. Then, the turntable was controlled to rotate at a constant speed of 2r/s. During this time, the fuze was reversed relatively to the turntable to keep the field of view stable. In addition, the turntable was controlled to perform random small-scale displacements in pitch and yaw directions while rotating. 

First, the stability of the mechanical system was verified without the pitch and yaw movement of the turntable. At the same time, the system did not run the AGGT. Figure 12a–c show the attitude of the fuze at three continuous moments during the mechanical stabilization and Figure 12d is the result of the azimuth. By comparing the position of the battery pack and the rudders, the fuze was basically in a relatively static condition. There only existed a small rotation due to the machining error of the components. It can be seen from Figure 11d that the error of the azimuth is only within 1 degree. The experimental results proved the effectiveness of the mechanical stabilization in this paper.

Based on the previous experiments, the system’s stabilization function was verified. To prove the robustness of the algorithm, the target was placed in the first and second quadrants and the turntable was controlled to perform random movement. The comparison of the azimuth before and after the electronic stabilization is shown in Figure 13, where the change in azimuth without electronic stabilization is indicated by the red curve and the green curve expresses the change of azimuth after stabilization. As shown, the accuracy of the azimuth was significantly improved after EIS and was closer to the true value. By calculating the variance of the azimuth, the numbers of the first quadrant was 9.308 degree and 3.348 degree with normal and stabilization conditions. Moreover, the numbers of the second quadrant was 12.14 degree and 4.365 degree. Therefore, an 64.1% accuracy increase was obtained. In addition, the two curves were basically coincident in the two plots, with only a few frames of deviation. This proved that the algorithm had a good online smoothing function. Therefore, the effectiveness of the strategy based on the AGGT was verified. 

However, two problems remain in the experimental data. First, an excessive stabilization is marked in Figure 12a. That is, the azimuth accuracy detection was lower than the normal value. By checking the experimental records many times, we found that the target had a trajectory phenomenon during the movement, which led to the wrong azimuth detection. This is a common phenomenon in infrared images during the high-frequency motion of the sensor. Moreover, the mark in Figure 12b shows the case of the missing frames and led to the two curves being misaligned from then on. This was due to the unstable exposure of the sensor. Obviously, it had little effect on the detection accuracy. 

## 5. Conclusions

A novel mechanical and electronic stabilization strategy based on infrared image sensor for trajectory correction fuze was proposed. The workflow of the mechanical stabilization was completed by a single-axis stepper motor and a small magnetometer sensor. Based on these designed structures and selected sensors, the fuze can basically remain relatively stationary with the rotating projectile. Then, the three main parameters of the target-based infrared image sensor were calculated. Additionally, the transformation relation between the projectile’s motion and the shaky video was investigated. Based on this, the EIS method was determined, in which the AGGT method and the gray projection algorithm were included. For two real infrared videos, our method produced more accurate motion vectors with a lower computational cost than the other three classical algorithms. Finally, the results show that the MIS can make the error of the fuze’s roll angle within 1 degree. Simulation results show that the average PSNR of the L1, Meshflow, Sobel and AGGT are 32.1dB, 20.5dB, 33.5dB and 35.1dB, respectively, in the first video. In the second video, the numbers are 27.3dB, 26.1dB, 30.2dB and 31.5dB, respectively. Compared with the L1, the Meshflow and Sobel methods, the average PSNR of the AGGT was improved by 4.3dB, 5.5dB and 1.4dB, respectively. Meanwhile, the average times of the Sobel and AGGT were 0.235s and 0.192s. The computational cost was reduced by 18.3%. A new mechanical and electronic experiment was implemented for further demonstration. The results show that an 64.1% accuracy increase was obtained for the target detection after the strategy was used. This research provides the theoretical and experimental basis for the application of sensors in an intelligent fuze.

As for the future work, we plan to combine video stabilization algorithms with dim target detection algorithms to further improve the efficiency. In addition, it would also be promising to implement the use of video stabilization algorithms on a variety of autonomous navigation ammunition. 

## Figures and Tables

**Figure 1 sensors-20-02461-f001:**
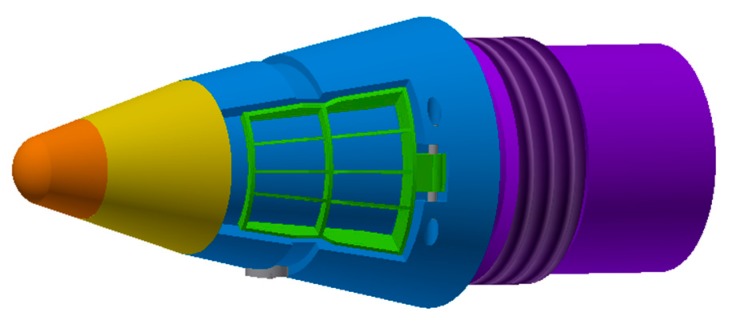
Appearance of the trajectory correction fuze.

**Figure 2 sensors-20-02461-f002:**
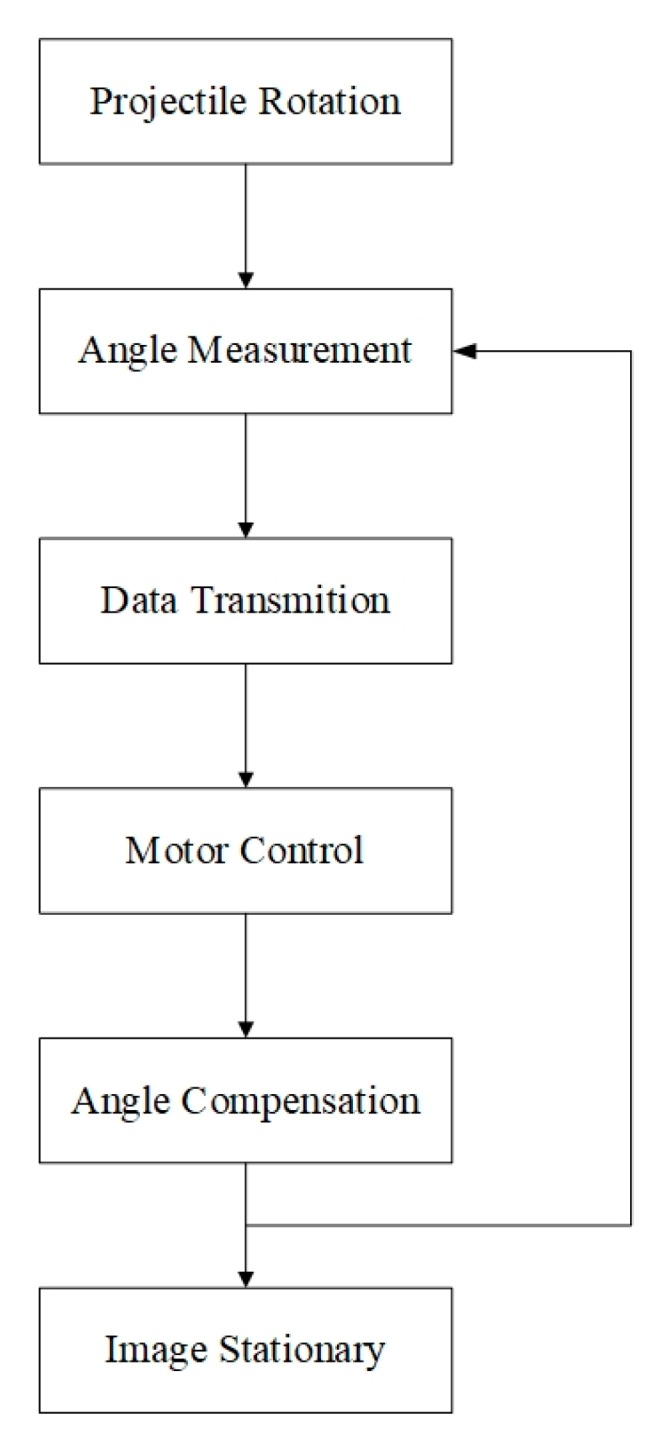
Workflow of the mechanical stabilization.

**Figure 3 sensors-20-02461-f003:**
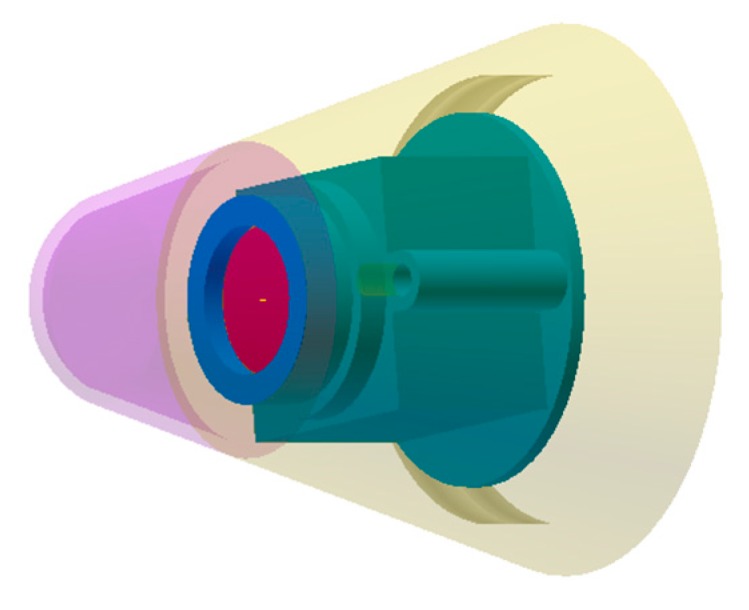
Composition of the infrared image system with the fuze.

**Figure 4 sensors-20-02461-f004:**
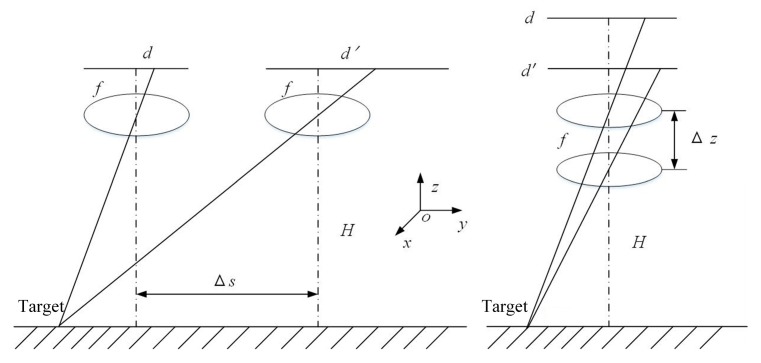
Pixel movement caused by translation.

**Figure 5 sensors-20-02461-f005:**
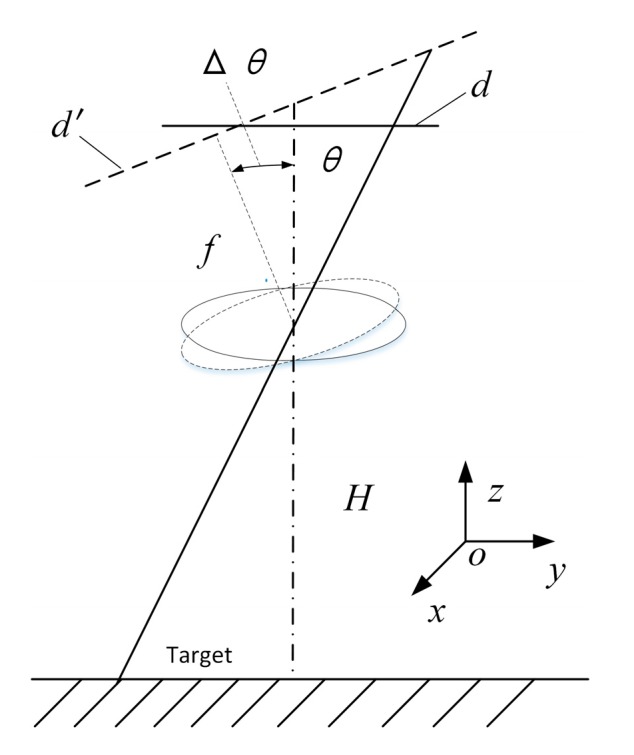
Pixel movement caused by shakiness.

**Figure 6 sensors-20-02461-f006:**
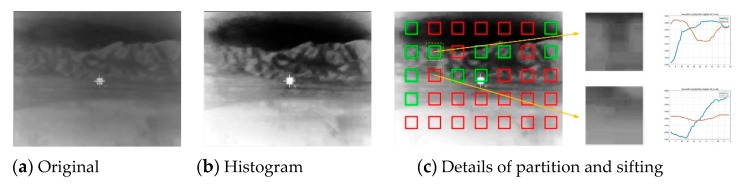
Partition and sifting sub-blocks.

**Figure 7 sensors-20-02461-f007:**
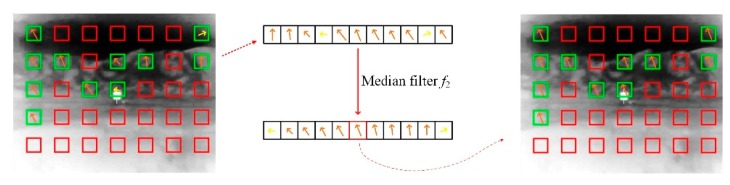
Median filter *f*_2_ to select the global motion vector.

**Figure 8 sensors-20-02461-f008:**
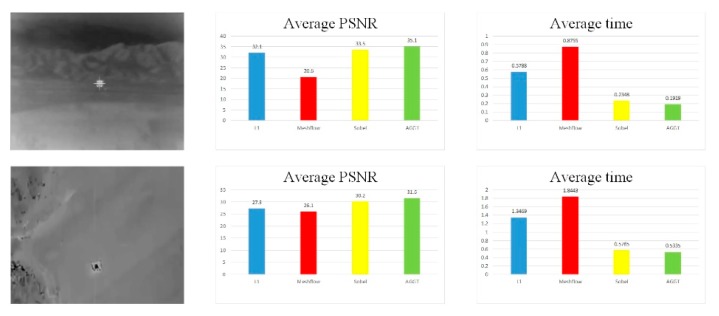
Comparison of two metrics.

**Figure 9 sensors-20-02461-f009:**
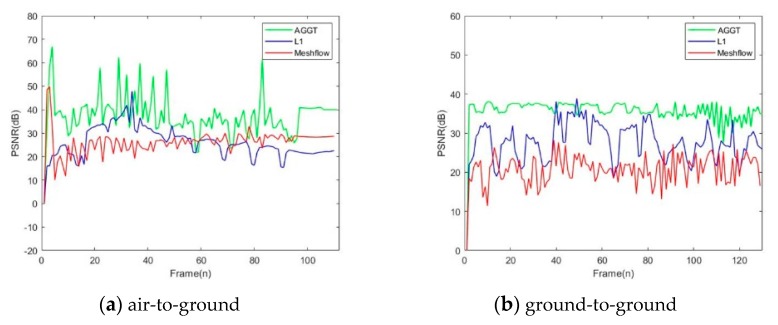
Comparison of the peak signal to noise ratio(PSNR).

**Figure 10 sensors-20-02461-f010:**
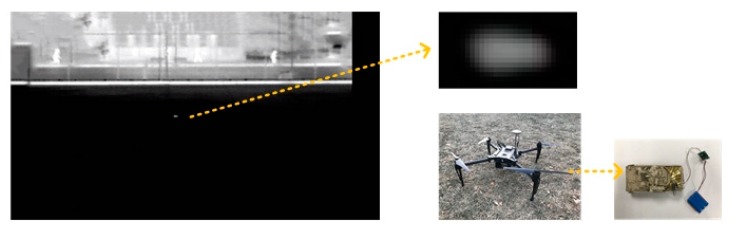
Experiment of the dim infrared target detection under the proportional scaling.

**Figure 11 sensors-20-02461-f011:**
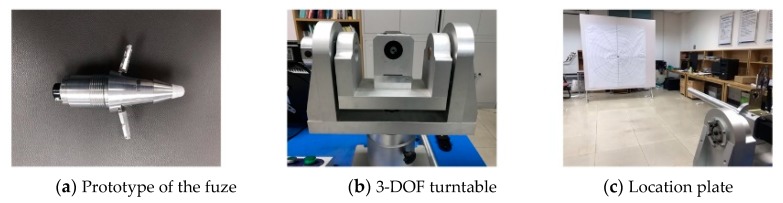
Experimental equipment and environment.

**Figure 12 sensors-20-02461-f012:**
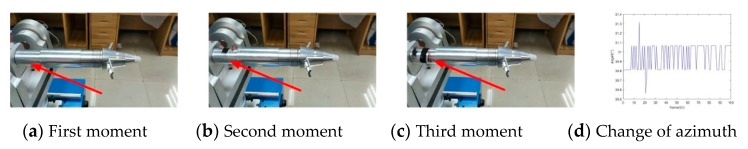
Experiment of mechanical video stabilization.

**Figure 13 sensors-20-02461-f013:**
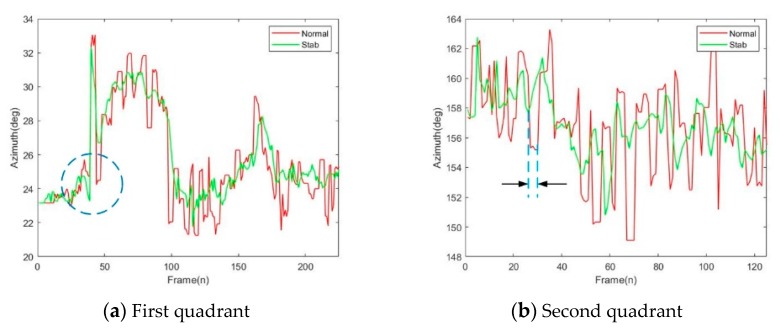
Experiment of the mechanical image stabilization (MIS) and the electronic image stabilization (EIS).

**Table 1 sensors-20-02461-t001:** Equivalent conditions and the parameters of the sensor.

Equivalent Conditions		Detector Parameters	
Distance (m)	65	Focal length (mm)	19
Model size (m)	0.1	Pixel size (um)	17
Pitch (degree)	53	FOV (degree)	17 × 13
Temperature (K)	298.15	Array format	320 × 240

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
