# Peer review of "Mechanical and Electronic Video Stabilization Strategy of Mortars with Trajectory Correction Fuze Based on Infrared Image Sensor"

_sensors, 2020, doi:10.3390/s20092461_

Round 1

Reviewer 1 Report

In this paper, the author proposed an approach for projectile trajectory correction fuze by using a combination of infrared image sensor and magnetometer sensor, together with some image processing algorithms. In general, the paper is well organized, the methods are properly described, and the results are validated by a lab-based experiment. However, there are two major concerns about the experiment: 1) the authors should perform a more systematic sensitivity analysis for the proposed approach in response to all the parameters listed in table; 2) the authors should consider some outdoor conditions (say high winds) which might affect the performance of the proposed approach, and how do you address them in this indoor experiment?

Reviewer 2 Report

The authors did a great job of presenting a new mechanical and electronic video stabilization strategy. The authors employ the correlation of the specified columns and rows to select images to process efficiently. Furthermore, the simulations and experimental results demonstrate the proposed strategy is feasible. No more comments can be provided.   

Author Response

Thank you very much for your review of the manuscript

Reviewer 3 Report

General: figures are a bit small only ussable for electronic publication.
Part of the paper reads as a commercial leaflet because not much data is given in the first pages only statements are made.
A lot of words without saying much. Could have been written much more compact.
It is more a story than a paper.

132 give a number for "high accuracy"
133 give "good real time abilty" a number
148,149 row subscript is missing, define m and n
214,215 number of pixels is reduced by factor 4 when only even blocks are used! Computational cost depend on algoritm ( proportional to #pixels or #pixels^2 or #pixels^(#pixels) ) is not defined. One can only state that the number of pixels is reduced.
216, 217 m missing, summation variable (right side) same as function variable (left side) they should be different.
249,250,251 one can only state that the number of pixel that needs to be processed is reduced with 69.4%.
252 the 82.7% not explained. no measurement or calculation given.
258 give a number for flight time,
260 give a number for frame rate
262 is not a correlation. It is variance. In a correlation formula 11 would be normalized with the prodcut of the sum of squares of cur and ref frames.
309 give the PSNR numbers and than the improvement.
355 10a should 11a
383 what is the actual accuracy give a number not only an improvement
407 give PSNR numbers not only improvements.
408 what was the computational cost give a number not only the improvement.

Round 2

Reviewer 1 Report

The authors have addressed my comments properly. It is now acceptable for publication.

Reviewer 3 Report

comments are addressed in the revised version.

Its ok now